

# Multi-objective test selection of smart contract and blockchain applications

Bader Alkhazi[1] and Amin Alipour[2]

[1] Kuwait University, Sabah Al-Salem University City, Kuwait
[2] University of Houston, Houston, United States of America

## ABSTRACT

The ability to create decentralized applications without the authority of a single entity has attracted numerous developers to build applications using blockchain technology. However, ensuring the correctness of such applications poses significant challenges, as it can result in financial losses or, even worse, a loss of user trust. Testing smart contracts introduces a unique set of challenges due to the additional restrictions and costs imposed by blockchain platforms during test case execution. Therefore, it remains uncertain whether testing techniques developed for traditional software can effectively be adapted to smart contracts. In this study, we propose a multi-objective test selection technique for smart contracts that aims to balance three objectives: time, coverage, and gas usage. We evaluated our approach using a comprehensive selection of real-world smart contracts and compared the results with various test selection methods employed in traditional software systems. Statistical analysis of our experiments, which utilized benchmark Solidity smart contract case studies, demonstrates that our approach significantly reduces the testing cost while still maintaining acceptable fault detection capabilities. This is in comparison to random search, mono-objective search, and the traditional re-testing method that does not employ heuristic search.

Corresponding author
Bader Alkhazi,
bader.alkhazi@ku.edu.kw

## INTRODUCTION

In recent years, blockchain (BC) technology has grown in popularity. This technology provides a computing paradigm that is decentralized and the immutable data structures afforded by it can address the trust in settings with multiple parties with no need of single authority. BC technology can offer several features such as immutability, decentralization, transparency, and security. Recent studies have shown successful application of the BC technology in healthcare, manufacturing, governance, and insurance domains (*Maesa & Mori, 2020*). These solutions are implemented by smart contracts (SC) that are self-executed computations that run on the nodes in the decentralized network. Despite the rapid adoption of the BC technology, there is a lack of tools and techniques to support the development and testing of the developed decentralized systems (*Lal & Marijan, 2021*; *Zou et al., 2019*). Specially, most SC are written in new languages (*e.g.*, Solidity) and decentralized platforms where most developers and college graduates lack formal training. In a study by *Zou et al. (2019)*, 54.7% of the interviewed developers reported a lack of specialized testing tools and the absence of practical testing guidelines as a problem

in development of safe and reliable SC. Such deficiencies could lead to development of incorrect and vulnerable SC, which in most scenarios can lead to extensive financial losses, such as a recent DAO attack where 3.6 million Ether (equivalent to $15 billion US dollars today) were stolen by the attacker (*Güçlütürk, 2018*).

Typically, most SC and decentralized applications (aka, DApps) use tools such as Truffle, Oyente (*Luu et al., 2016*), and Contractfuzzer (*Jiang, Liu & Chan, 2018*) for testing. However, it's important to note that all testing is conducted in a test environment. While early testing is considered an effective quality assurance strategy for standard software systems, BC applications can be impacted by external factors that cannot be accurately predicted or simulated in a test environment. These factors include network congestion, consensus mechanisms, the number of available miners, end-users' preferences (*i.e.,* GEWI, gas limit, wallet), and other BC networks that the contract is designed to interact with. Validating the performance of BC applications sometimes requires testing in both controlled test environments and real-world scenarios. According to *Chainalysis (2022)*, 13 separate attacks on bridges have resulted in the theft of approximately $2 billion worth of tokens . It is not necessarily accurate to say that the victims of these attacks failed to perform adequate testing before deploying their bridges. In fact, even the largest exchange, Binance, suffered a loss of $570 million due to a bridge incident last year (*Shukla & Irrera, 2022*). The complexity of these tools, compounded by the fact that different blockchains are often written in distinct programming languages and deployed in varying virtual environments, can make it extremely challenging to determine how they should interact. While testing in a controlled environment is important, additional testing on a live network can provide valuable insights and benefits. However, testing on a live network should be approached with caution, as it may expose potential security vulnerabilities or other issues to the public. Therefore, appropriate precautions should be taken during live testing. It is worth noting that live testing should not replace testing on a controlled environment, but rather complement it. Nevertheless, it should be considered that using the live network for testing comes with a cost per transaction, which can make it impractical and expensive to run all test cases again. Therefore, test selection in BC is critical. Unfortunately, unlike mainstream software systems where there are numerous studies in test selection and optimization (*Yoo & Harman, 2012*), we cannot say the same for the SC applications.

In this study, we aim to formulate the test case selection as a multi-objective optimization problem to find the most efficient test suite that meets the following conditions: (1) minimize execution time: transactions on the Ethereum network requires time, unlike local networks, it may take anywhere between 15 and 300 s. Theoretically, the network does not have a timeout, therefore, a transaction can last forever. Thus, reducing the execution time on a local network will minimize the possibility of spending 10x the time on the actual platform. The second objective is to (2) minimize gas cost that aims to reduce the monetary cost of test execution. In fact, performing operations on the Ethereum BC involves paying fees to the miners and to keep bad actors away from flooding the network with denial of service attacks. Therefore, it is important to keep the testing activities within a reasonable budget without compromising the third objective which is to (3) maximize coverage. Code coverage has traditionally been used as as a metric in quality of test suite. While high code

coverage cannot guarantee correctness of software, lack of it poses serious doubts about the quality of tests.

We evaluated the effectiveness of the proposed approach based on five real-world Solidity projects from State of the DApps (https://www.stateofthedapps.com) and GitHub. We compared the effectiveness of our approach against other common test selection techniques including mono-objective test selection and baseline random selection. The main contributions of this paper are:

1. The paper presents the first study for SC test case selection. To address the competing objectives of coverage and cost, we used a multi-objective algorithm to select the test cases in order to minimize gas cost and execution time while maximizing the coverage.
2. The paper compares the effectiveness of the proposed approach with the state-of-the-art approaches based on real-world Solidity projects.
3. The paper provides a set of guidelines for test case selection in SC by analyzing the selected test cases chosen by the approach.

The remainder of this paper is structured as follows. We first introduce necessary background and related work in 'Background and Related Work'. Our approach is described in 'Test Cases Selection for smart Contracts'. Evaluation including execution and analysis plans are in 'Validation'. Finally, threats to validity and conclusion are presented in 'Threats to Validity and Conclusion', respectively.

# BACKGROUND AND RELATED WORK

## Background

In this section, we provide a brief overview of the main concepts in block chain technology and SC.

### Blockchain technology

A BC is a decentralized and distributed ledger consisting of chronically ordered blocks where each block consists of a series of transactions (*Nakamoto, 2008*). In order to ensure the immutability of the data stored in the ledger, every block contains a hash of its predecessor. Since BC follows a P2P topology, the ledger usually has duplicates in different nodes across the network. Therefore, to append new blocks to the ledger, a consensus mechanism needs to be performed. In short, the consensus mechanism is a protocol, *e.g.*, Proof-of-Work (PoW), and Proof-of-Stake (PoS), that defines the conditions required for a new block to be added to the BC (*Xiao et al., 2020*). Therefore, nodes across the network can achieve the necessary agreement on new data values before updating the ledger.

BC can be classified into two categories: permissioned or permissionless. The former is used for private networks, where only pre-defined nodes are allowed to join the network. The permissionless type, however, is public. Therefore, anyone can take part in the consensus process, store a ledger state, send and receive transactions, *etc*. The type of BC is decided based on its objective. If the application requires an open and accessible environment, transparency of transactions, or the absence of a central authority, a permissionless BC may be the preferred choice. On the other hand, if privacy is important, the speed of transactions is vital, or when customization is required, a private

(permissioned) BC is a better fit. Overall, BC-based systems provide valuable and unique features such as decentralization, immutability, provenance, and consensus making it the perfect solution to many of our modern technical difficulties (*Wu et al., 2019b*; *Maesa & Mori, 2020*).

### Smart contracts

SC are programs stored on a BC (*e.g.*, Ethereum), which run when certain conditions are met. Since transactions are irreversible and traceable, SC is broadly used to handle transactions between disparate parties without the supervision of a central authority or the enforcement of a legal system (*Buterin, 2014*; *Macrinici, Cartofeanu & Gao, 2018*). SC typically are written in special high-level programming languages such as Solidity and Go Lang. The former, however, is the most popular language used especially with contracts deployed to the Ethereum platform (*Dannen, 2017*). SC programs usually consist of three main structures: sequential, selection, and loop structure (*Grishchenko, Maffei & Schneidewind, 2018*). Solidity has similar coding fundamentals to common programming languages such as C++, Python, and JavaScript. Therefore, it supports inheritance between SC, libraries, loops, and user-defined types. Besides its contract-oriented programming capabilities, Solidity has distinct properties that are tailored for writing code on the Ethereum BC. For instance, developers can use "require" statements in Solidity to set conditions that must be fulfilled for a function to execute. Additionally, the "payable" function modifier lets functions receive and transfer Ether. Furthermore, Solidity provides the "self-destruct" function, which allows developers to remove a contract and transfer its remaining Ether balance to another account.

```solidity
1  pragma solidity ^0.6.0;
2
3  contract MembershipSubscription {
4    uint256 monthlyFees;
5
6      constructor(uint256 fee) public{
7          monthlyFees = fee;
8      }
9
10     function makePayment() payable public {
11     }
12
13     function isBalanceValid(uint256 monthsElapsed) public view returns (bool) {
14         return monthlyFees * monthsElapsed >= address(this).balance;
15     }
16
17     function withdrawBalance() public {
18         msg.sender.transfer(address(this).balance);
19     }
20 }
```

Listing 1: Sample Smart contract

A sample SC code is shown in Listing 1. The first line defines compiler compatibility. In this case, the code is compatible with Solidity version 0.6.0 onwards. Line 3 sets the name of the contract as (MembershipSubscription), which is the name used in our example.

Line 4 defines monthlyFees as an unsigned integer of 256 bits. Next, the constructor is defined in line 6. In Solidity, the constructor is only called once after deployment, therefore

it cannot be called explicitly afterwards. The constructor in our example initializes the `monthlyFees` variable with the passed argument `fee` (line 7). The function `makePayment` in line 10 allows the customer to pay their fees. By default, Ethereum SC act as wallet. This allows the SC to receive, store, and send tokens and digital coins. The body of this function is empty as the term `payable` allows the SC to automatically store the sent value internally to be used later by other functions. On line 13, the function `isBalanceValid` takes `monthsElapsed` as a parameter and returns true only if the balance of this account is enough to pay the bill. The `view` keyword indicates that this function will not change the internal state (read-only). Finally, the function `withdrawBalance` in line 17 sends the entire account balance to the recent caller. Note that in real world contracts, a restricted pre-approved list of sender addresses will be allowed to withdraw the balance to prevent random users from stealing the funds. However, for simplicity purposes, we made the example as basic as possible. In the following subsection, we present an overview of software testing for traditional and BC applications including SC.

### Software testing

Software testing is a set of activities carried out either manually or automatically to identify the correctness of the software and detect bugs. While the testing process may find errors in a system, passing all test cases does not guarantee their absence. Since exhaustive testing for complex systems is practically infeasible, there are two main testing approaches: white box and black box testing. In white box testing, the internal code structure is known to the tester when designing the test cases. This approach is mainly used for unit testing. In black box testing, the behavior or the functionality of the system is the focus, therefore knowing much about the internal perspective of the system is not needed. Thus, this approach is often called Input-Output testing. The types of testing can be further classified as unit, integration, system, and acceptance testing (*Lal & Marijan, 2021*). The objective of the unit testing is to ensure that individual parts of the code work flawlessly. After combining several code components together, an integration test is performed to confirm that these units are coordinating correctly. The next step is system testing, where the aim is to check that the entire system is working as stated in the requirements. The final step is to test whether the customer will find the system good enough to be accepted and delivered to its intended users. The complexity and cost of testing increases as the project progresses; therefore, it is encouraged to perform testing as early as possible to fix bugs before propagating across the entire system.

## Related work
### Test case selection and prioritization

Testing activities consumes up to 50% of the project's development cost (*Singh & Singh, 2012*), therefore, optimizing the test suite will have a positive impact on the budget and the quality of the software since developers will have more time to build and debug the code. There are many techniques used to improve the efficiency of software testing such as test case selection, reduction, and prioritization (*Yoo & Harman, 2012*). Test case prioritization is the process of arranging test cases based on particular criteria to make software testing more effective (*Elbaum, Malishevsky & Rothermel, 2002*). Test case selection, however,

achieves the same objective by selecting a subset of the available test cases based on certain preferences.

Integer programming was employed in early work on test case selection methods such as the work in *Fischer (1977)*, *Fischer, Raji & Chruscicki (1981)* and *Lee & He (1990)*. *Hartmann & Robson (1989)* and *Hartmann & Robson (1990)*, extended this work to be used for C programs. Heuristics have been employed in some studies to choose test cases; in *Biswas et al. (2009)*, the authors used genetic algorithms. While in *Mirarab, Akhlaghi & Tahvildari (2012)*, *Kumar, Sharma & Kumar (2012)*, *Panichella et al. (2015)*, *de Souza, Prudêncio & Barros (2014)*, *Yoo & Harman (2007)* and *Alkhazi et al. (2020a)*, the researchers adopted multi-objective optimization methods to choose the suitable cases.

Recent studies used evolutionary algorithms to solve the test case prioritization problem (*Li, Harman & Hierons, 2007*). In *Tulasiraman & Kalimuthu (2018)*, the authors used historical information of test cases such as the severity of fault identified and execution time in order to prioritize test suites using a clonal selection algorithm. The authors in *Konsaard & Ramingwong (2015)* addressed the problem of test case prioritization by proposing a modified adaptive genetic algorithm that ranks test cases based on their total coverage. When compared to five other algorithms, their approach yielded better results. *Khanna et al. (2018)* applied multi-objective algorithms on web testing to prioritize test cases based on execution time and fault detection. The proposed approach achieved better results than random search, greedy algorithm, and weighted genetic algorithm. In *Yadav & Dutta (2017)*, the Average Percentage of Statements Covered (APSC) was used as an evaluation metric for a genetic algorithm, the study delivered improved results when compared with other techniques. Multiple coverage metrics were used by *Ahmed, Shaheen & Kosba (2012)* to prioritize test cases. They defined three fitness functions to evaluate the proposed solutions. The three metrics were (1) condition coverage, (2) multiple condition coverage, and (3) statement coverage. The authors evaluated the performance of their proposed work using APFD, and results showed a significant improvement in the average percentage of fault detected. Other studies tackled the prioritization problem using different algorithms and approaches such as Ant Colony Optimization (ACO) (*Panwar et al., 2018*), Bayesian networks (BN) (*Mirarab & Tahvildari, 2007*), and neural networks (*Gökçe & Eminli, 2014*; *Gökçe, Eminov & Belli, 2006*). Given the abundance of research in this area, we recommend the following survey papers for the interested readers: (*Yoo & Harman, 2012*; *Biswas et al., 2011*; *Rosero, Gómez & Rodríguez, 2016*; *Kazmi et al., 2017*).

Our proposed work is categorized with test suite selection solutions using a multi-objective approach. However, the used software artifact (*i.e.,* SC) is a new application domain for these techniques.

### Smart contract testing

The majority of existing SC analysis tools focus on vulnerability detection. Some of these tools use static program analysis such as GASPER (*Chen et al., 2017*), Teether (*Krupp & Rossow, 2018*), Oyente (*Luu et al., 2016*), Securify (*Tsankov et al., 2018*), MAIAN (*Nikolić et al., 2018*), and Smartcheck (*Tikhomirov et al., 2018*). A random fuzzing is used to find security issues in ContractFuzzer (*Jiang, Liu & Chan, 2018*), while *Su et al. (2022)*

introduced RLF, a reinforcement learning-based vulnerability-guided fuzz testing approach for detecting complex vulnerabilities in SC. They modeled the fuzz testing process as a Markov decision process, with a reward that accounted for vulnerabilities and code coverage. This approach effectively generated transaction sequences to reveal vulnerabilities, particularly those related to multiple functions. The authors in *Li et al. (2019)* proposed MuSC: a graphical user interface that performs mutation operations at the level of abstract syntax tree. The work of *Grishchenko, Maffei & Schneidewind (2018)*, Zeus (*Kalra et al., 2018*), and Vandal (*Brent et al., 2018*) utilizes formal verification tools in their approaches.

Recently, there have been several developments in test case generation approaches for SC. Similar to ContractFuzzer, the authors in *Akca, Rajan & Peng (2019)* proposed SolAnalyser that randomly generates test cases to detect vulnerabilities. More complicated test case generator (sfuzz) are proposed in *Nguyen et al. (2020)*. In their proposed approach, they used a search-based technique to generate test cases that focus on uncovered branches. Similarly, *Driessen et al. (2021)* proposed AGSOLT. Their automatic test suite generator composed of two stages: the initialization phase and testing loop phase. In the first step, the tool extracts relevant data from the SC. In the second phase, however, the test cases are randomly generated, executed on a local testnet (Ganache), and their performance is measured to decide whether they should move to the next iteration or not. The latter step uses genetic algorithm to iteratively improve the initially generated random test cases for better branch coverage. *Wang et al. (2019)* also tackled the efficient generation of test cases for SC using genetic algorithms. They validated their approach on 8 SC to show that branch coverage improved compared to random generation. Additional tools that took advantage of mutation testing include Regularmutator (*Ivanova & Khritankov, 2020*; *Andesta, Faghih & Fooladgar, 2020*, and *Hartel & Schumi (2020)*).

To summarize, while many researchers investigated test case generation for SC, the problem of efficiently and effectively testing SC has mostly been ignored. To the best of our knowledge, our proposed work is the first test case selection approach for smart contacts.

# TEST CASES SELECTION FOR SMART CONTRACTS

In this section, we describe our multi-objective approach for test case selection for SC using a search-based technique, namely Non-Dominated Sorting Genetic Algorithm II (NSGA-II).

## Approach overview

The main objective of our approach is to optimize a test suite to achieve three goals as illustrated in Fig. 1. Initially, we take the SC in addition to its existing test cases as inputs. Our algorithm will begin by analyzing and collecting certain data such as coverage, execution time and cost, test case size, and so on. Our objective is to maximize coverage, reduce total execution time, and minimize the gas cost. Since these objectives are conflicting, and because we are inherently dealing with a huge search space, we will utilize a multi-objective algorithm to locate the Pareto-optimal solutions for this problem. An overview of this algorithm and how it can be used in our case will be illustrated in the subsequent section.

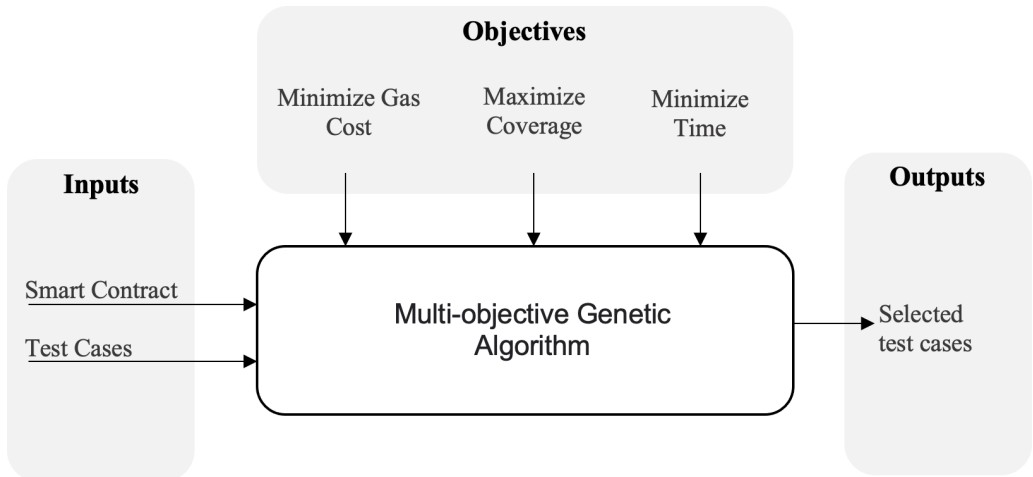

**Figure 1** An overview of the proposed approach.

## Search-based formulation

---

**Algorithm 1** Pseudo code of NSGA-II adaptation for smart contract test-cases prioritization

---

1: **Inputs:** Solidity smart contract $P$, Test suite $TC$
2: **Output:** subset(s) of the test suite
3: **Begin**
4:   I:= Instantiation($TC$)// vectors of TCs
5:   $P_0$:= set_of(I)
6:   t:= 0
7: **Repeat**
8:   $C_t$:= apply_Genetic_Operators($P_t$)
9:   $G_t$:= $P_t \cup C_t$ // Combine parent and offspring populations
10: **for all** I ∈ $G_t$ **do**
11:     Execution_Time(I):= calculate_Execution_Time($P$)
12:     Coverage(I):= calculate_Code_Coverage($P$)
13:     Cost(I):= calculate_Execution_Cost($P$)
14: **end for**
15: F:= fast_Non_Dominated_Sort($G_t$) // F=($F_1, F_2, \ldots$), all nondominated fronts of $G_t$
16: $P_{t+1} = \varnothing$
17: i:= 1
18: **while** $|P_{t+1}| + |F_i| <$ Max_size **do**
19:     Crowding_distance_assignment($F_i$) // calculate crowding distance in $F_i$
20:     $P_{t+1}$= $P_{t+1} \cup F_i$ // include $i$th nondominated front in parent pop
21:     i:= i+1
22: **end while**
23: Sort ($F_i, \prec_n$) // sort in descending order using $\prec_n$
24: $P_{t+1}$= $P_{t+1} \cup F_i [1\ldots(Max\_size - |P_{t+1}|)]$ // choose the first Max_size $- |P_{t+1}|$ elements of $F_i$

25: t:= t+1 // increment generation counter
26: **until** t=Max_iteration
27: best_solutions := first_front($P_t$)
28: **return** best_solutions

---

In the literature (*Mkaouer et al., 2015*; *Alkhazi et al., 2020b*), the multi-objective problem as follows is represented by the following formula:

$$\begin{cases} Min f(x) = [f_1(x), f_1(x), \ldots, f_M(x)]^T \\ g_j(x) \geq 0 & j = 1, \ldots, P; \\ h_k(x) = 0 & k = 1, \ldots, Q; \\ x_i^L \leq x_i \leq x_i^U & i = 1, \ldots, n; \end{cases}$$

In short, the number of objective functions is represented by $M$. Whereas $Q$ and $P$ are the equality constraints and inequality constraints, respectively. The lower bound of the decision variable is represented by $x_i^L$, and the higher ones by $x_i^U$.

Each solution $x$ is represented by a number of decision variables. The metaheuristic algorithm objective is to optimize these variables. $\Omega$ is a set of all feasible solutions (search space), which are the solutions that met the constraints $((P + Q))$. To calculate the objective value for a particular solution $f_i$, the *fitness function f* is evaluated where all objectives should be minimized. When the maximum value of an objective is desired, we simply take its negative value to meet the algorithm's condition. A high-level pseudo-code of NSGA-II is shown in Algorithm 1.

## Pareto-optimal solutions

For each multi-objective problem, we evaluate its defined objective functions for a specific solution. By comparing the objective vectors of two solutions, we can figure out which one is 'better' based on these objectives. One common way to do this comparison is by adding up all the objective values of one solution and comparing the total with that of another solution. However, this only works if all the values are in the same units of measurement. In the context of SBSE, we frequently rely on the concept of Pareto optimality. As defined in (1) and (2) with strict inequality *(Harman, 2007)*, Pareto optimality means that a solution is considered superior to another if it outperforms it in at least one objective function and doesn't perform worse in any other. This definition helps us identify which solution is better, although it doesn't provide a measure of 'how much' better it is.

$$F(\overline{x1}) \geqslant F(\overline{x2}) \Longleftrightarrow \forall_i f_i(\overline{x1}) \geqslant f_i(\overline{x2}) \tag{1}$$

$$F(\overline{x1}) > F(\overline{x2}) \Longleftrightarrow \forall_i f_i(\overline{x1}) \geqslant f_i(\overline{x2}) \wedge \exists_i f_i(\overline{x1}) > f_i(\overline{x2}) \tag{2}$$

In the field of SBSE, the algorithms utilize the concept of Pareto optimality as part of the search process to generate a collection of non-dominated solutions. Each of these non-dominated solutions can be thought of as a balanced trade-off across all objective functions, where no solution in the set is definitively better or worse than another. It's important to acknowledge that SBSE operates under the assumption that accurately determining the 'true' Pareto front of a problem—which is the set of all values that are Pareto optimal—is analytically infeasible and impractical to achieve through exhaustive search. Consequently, every set produced through metaheuristic search serves as an approximation of this often elusive 'true' Pareto front (Fig. 2). Further iterations of such an algorithm could potentially enhance this approximation.

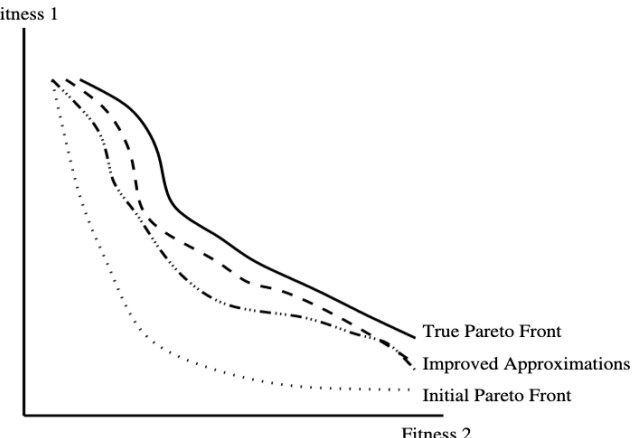

**Figure 2 Pareto optimality and pareto fronts.** From *Harman (2007)*.

**Table 1 Sample solution representation.**

| | |
|---|---|
| 1 | Test_Case(22, 32, 533.1, Statement[S7,S1,S5,S7,S7]) |
| 2 | Test_Case(15, 18, 302.04, Statement[S1,S4]) |
| 3 | Test_Case(6, 89, 941.93, Statement[S10,S9,S3,S7,S8,S2]) |
| ... | .... |
| NN | Test_Case(11, 105, 211.7, Statement[S10,S9,S3,S2]) |

## Solution representations

Table 1 is a sample solution vector. Each solution is represented by a vector where every dimension constitutes a test case. After analyzing each test case to measure its execution time, gas cost, and coverage, we use this data in the solution representation as shown in Table 1 where each test case constitutes: Test_Case(*ID, Gas Cost, Execution Time, and Covered Statements.*)

```solidity
// SPDX-License-Identifier: GPL-3.0
pragma solidity ^0.8.4;

contract donations{
    struct Donation {
        uint id;
        uint amount;
        string donor;
        uint timestamp; //seconds since unix start
    }
    uint amount = 0;
    uint id = 0;
    mapping(address => uint) public balances;
    mapping(address => Donation[]) public donationsMap;

    function donate(address _recipient, string memory _donor) public payable {
        require(msg.value > 0, "The donation needs to be >0 in order for it to go
        through");
        amount = msg.value;
        balances[_recipient] += amount;
```

```
20          donationsMap[_recipient].push(Donation(id++,amount,_donor,block.timestamp)
        );
21      }

23      function withdraw() public {   //whole thing by default.
24          amount = balances[msg.sender];
25          balances[msg.sender] -= amount;
26          require(amount > 0, "Your current balance is 0");
27          (bool success,) = msg.sender.call{value:amount}("");
28          if(!success){
29              revert();
30          }
31      }

32
33      function balances_getter(address _recipient) public view returns (uint){
34              return balances[_recipient];
35      }

36
37      function getBalance() public view returns(uint) {
38              return msg.sender.balance;
39      }
40 }
```

Listing 2: Sample smart contract adopted from Remix documentation Remix (2021)

```
1 // SPDX-License-Identifier: GPL-3.0
2 pragma solidity ^0.8.4;
3 import "./donations.sol";

5 contract testSuite is donations {
6     address sender = TestsAccounts.getAccount(0);
7     address recipient = TestsAccounts.getAccount(1);

9     function donateAndCheckBalance() public payable{
10        Assert.equal(msg.value, 1000000000000000000, 'value should be 1 Eth');
11        donate(recipient, "Bader");
12        Assert.equal(balances_getter(recipient), 1000000000000000000, 'balances
        should be 1 Eth');
13    }

15 }
```

Listing 3: Sample test case for the contract in listing 2

A simple Solidity contract is shown in listing 2, and a sample test case for this contract is shown in listing 3. Executing this test case will cover 50% of the statements leaving half of the functions untested. While executing this single test case required only 0.86 s on a local test BC, executing a transaction on the mainnet requires 15 s on average. The speed of the transaction on the mainnet is affected by different factors such as network congestion, paid gas, and number of available miners. The consumed gas for this single test case was 21,484. Therefore, executing this test case on the mainnet will cost around $4.0 at the current average gas price (4~6 Gwei), making running more cases a burden on the testing team.

In general, the initial population is selected randomly, and the vector $V$ is bound by a maximum size $V_{MAX}$ that is proportional to the SC's statements and the test cases available for selection.

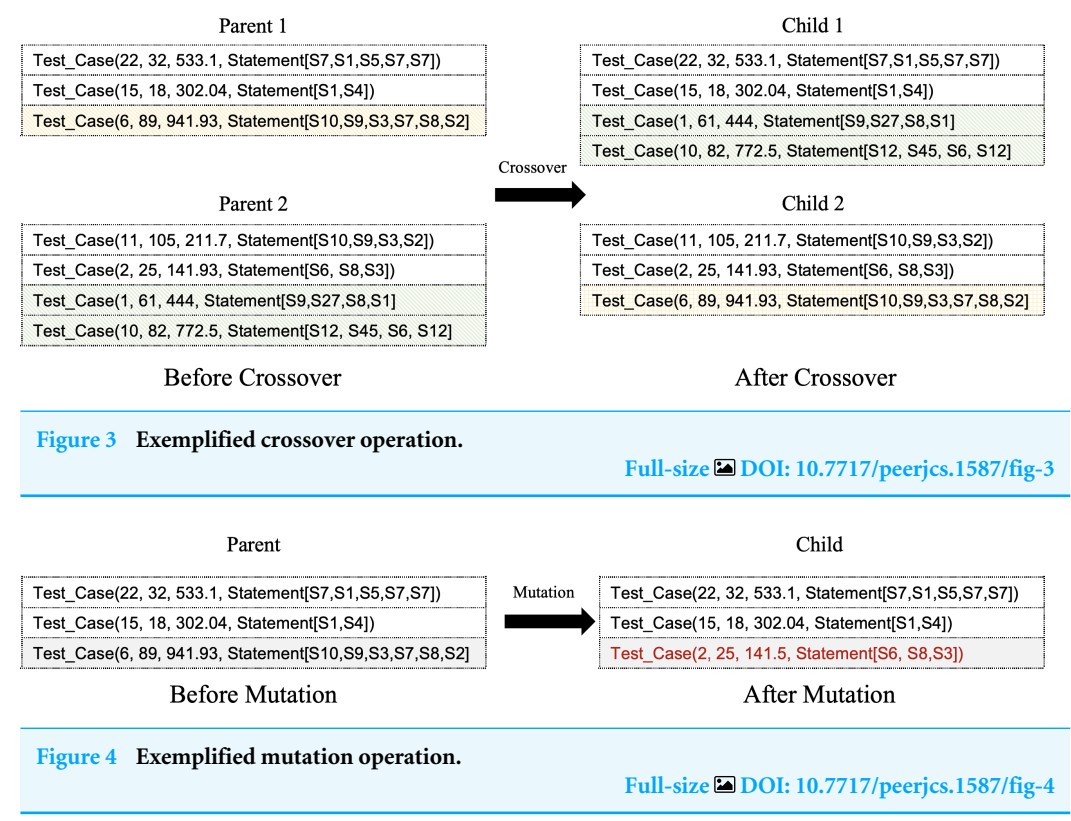

**Figure 3** Exemplified crossover operation.

**Figure 4** Exemplified mutation operation.

## Solution variation

Variation operations are vital in exploring the search space. They allow the algorithm to potentially find better alternative solutions by using the crossover, mutation, and selection of the fittest. In our case, we will use a one-point crossover operation where the parent solutions are split at an arbitrary location. The next step is crossing the split parts between the parents to form new children as shown in Fig. 3. The crossover allows for better convergence in a subspace. The mutation, however, helps to diversify the population and provides a way to escape a local optimum. In our study, we use the bit-string mutation operator to randomly select one test case in the vector dimension in order to replace it with another entry. Figure 4 illustrates a high-level overview of the mutation operator.

## Evaluation metrics

During this step, we make sure that we select the fittest solutions (elitism) to be carried out for the next iteration. Therefore, we use three fitness functions to evaluate and rank solutions to make a better decision about whether a solution should pass to the next generation or not.

**Lower gas cost**: To perform an operation on the Ethereum BC (or almost any BC), a particular amount of gas needs to be paid. The motivation for this fee is twofold. First, the BC will be free from bad actors and spammers who may flood the BC to cause congestion and breakdowns. This is because the transaction originator should specify the amount of Ether they are willing to pay during the life cycle of their transactions. Therefore, bad actors need a significant amount of capital to flood the network. The same applies to poor

programming practices such as infinite loops, where the transaction would eventually fail and throw an out-of-gas exception. The second reason for the gas fees is to pay the miners for processing the transactions and including it in the block. The transaction cost is determined by two factors: amount of gas used, and the gas price as shown in the following formula:

$$GasCost = Used\_Gas * Gas\_Price \tag{3}$$

The goal is to organize the test cases in a way where testers who are on budget can run a larger number of test cases. It is worth mentioning that gas price varies over time based on the number of available miners, network congestion, and Ether price.

**Maximum coverage:** The objective is simply to increase the SC coverage. After executing each test case, we trace the triggered code statements by that case. We later combine the aggregated covered statements to measure the coverage score. It is important to note that *coverage and gas costs are not correlated.* In solidity, some statements and functions will cost lower gas than others since they do not access the BC state. For instance, *Pure* functions in Solidity are purely computational and deterministic. They only operate on input parameters to compute a value and do not modify the state of the BC. As a result, they are less expensive in terms of gas than functions that do modify the state. Thus, covering these functions in test cases can improve code coverage without significantly increasing gas consumption.

**Lower execution time**: A shorter execution time is desired; therefore, we aim to minimize the time needed to discover faults. We will use a test net to run every test case and record its execution time. This data will be used in the following step to prioritize test solutions with lower execution time.

## VALIDATION

### Research Questions

To evaluate our approach, we defined three research questions as follows:

- **RQ1: Search validation (sanity check).** To assess the need of our problem formulation, we compared our multi-objective approach to a random search algorithm (RS). If RS proves to be more effective than a sophisticated search method, it would indicate that there is no need for the use of metaheuristic search.
- **RQ2: How efficient is the proposed multi-objective method in comparison to mono-objective approach?** To confirm that the objectives are in opposition, we combined the three normalized objectives into a single fitness function. If the results are equal or the mono-objective approach outperforms the multi-objective approach, it suggests that the latter is unnecessary. The single-objective genetic algorithm, which has the same design as NSGA-II but only for optimizing a single objective, was used for the mono-objective approach. It does not utilize the non-dominance principles and crowding distance in generating the Pareto front.
- **RQ3: How effective is our multi-objective test case selection method in revealing faults?** A well-designed test selection approach plays a vital role in ensuring the success

of a software system by striking a balance between cost and time efficiency and thorough fault detection. By carefully selecting the tests to be executed, this approach minimizes the cost and time needed to run the tests, while preserving the test suite's effectiveness in uncovering faults, thereby maintaining the validity of the software's functionality and performance.

## Case Studies

To evaluate the research questions, we used five solidity SC case studies based on data collected from GitHub and State of the DApps (https://www.stateofthedapps.com). We only considered projects that are hosted in the top five BC platforms, written in Solidity, and has its open-source code for the SC published. We did not include stealth, inactive, or beta projects. In addition, projects with less than 100 lines of code for their combined SC was not considered for evaluation.

The case studies employed vary in size, application area, and structure. The solidity programs that we used are briefly described in the section below.

- **Chainlink** is a decentralized oracle network that connects SC to real-world data, events, and payments. It enables SC to securely access off-chain data feeds, web APIs, and traditional bank payments. This allows for the creation of more advanced and reliable SC, which can be used for a wide range of purposes such as financial applications, supply chain management, and gaming. The Chainlink network is composed of a network of independent, security-reviewed node operators that provide data to SC, and is secured by a decentralized network of nodes. The token of Chainlink (LINK) is used as a form of payment to the node operators for providing these services. The project is open-source and is supported by a large and active community of developers and users.

- **PancakeSwap** is a decentralized exchange (DEX) built on the Binance Smart Chain (BSC) that allows users to trade various tokens built on the Binance ecosystem. The exchange utilizes an automated market maker (AMM) model, where users can trade tokens without the need for a traditional order book. Additionally, PancakeSwap features a unique liquidity mining mechanism, where users can provide liquidity to the exchange and earn rewards in the form of CAKE tokens. This incentivizes users to provide liquidity to the exchange, increasing its overall liquidity and contributing to its overall decentralization. The platform is fully decentralized and is governed by its community of users through a decentralized autonomous organization (DAO). PancakeSwap has quickly gained popularity among the decentralized finance (DeFi) community for its user-friendly interface, low transaction fees, and high liquidity.

- **Venus Protocol** The Venus Protocol is a decentralized finance (DeFi) platform built on the Binance Smart Chain. It aims to provide a stablecoin-based yield farming and lending platform, where users can lend and borrow assets, earn interest, and participate in liquidity provision. The Venus Protocol uses a collateralized stablecoin, called VAI, as its base currency, which is pegged to the value of the US dollar. This allows for more stable lending and borrowing rates, as well as reducing the volatility of returns for yield farming. Additionally, the Venus Protocol features a unique governance system, where users can vote on and propose changes to the protocol using their XVS tokens. This

allows for a more decentralized and community-driven approach to the development and management of the protocol. The Venus Protocol aims to provide a fair, transparent, and sustainable DeFi platform for users to earn returns on their assets and participate in the growth of the ecosystem.

- **Axie Infinity** is a play-to-earn decentralized gaming platform built on the Ethereum BC. It allows players to earn cryptocurrency by participating in the game, such as by breeding valuable Axies or participating in in-game events. The game features unique collectible creatures called Axies that can be bought, bred, and battled. The genetic system allows players to breed different Axies and create new and unique offspring. It also has a marketplace where players can buy, sell and trade different Axies. The platform also features a unique governance system where players can vote on and propose changes to the game using their SLP tokens. The game has a vibrant community and is actively developed by the team behind it.

- **SushiSwap** is a decentralized exchange (DEX) built on the Ethereum BC that allows users to trade various tokens, including Ethereum and other ERC-20 tokens. The exchange utilizes an automated market maker (AMM) model, where users can trade tokens without the need for a traditional order book. It also features a unique liquidity mining mechanism, where users can provide liquidity to the exchange and earn rewards in the form of SUSHI tokens. This incentivizes users to provide liquidity to the exchange, increasing its overall liquidity and contributing to its overall decentralization. SushiSwap also supports yield farming, where users can earn additional rewards by providing liquidity to certain pools. The platform is fully decentralized and is governed by its community of users through a decentralized autonomous organization (DAO). SushiSwap gained popularity among the decentralized finance (DeFi) community for its unique features, low transaction fees, and high liquidity.

Table 2 summarizes the structural details of the test cases we used in our study. The table shows the number of SC for each project, the total number of functions and branches, and the total number of lines of code (LoC). In Table 3, however, we describe the test cases available for selection for each project, the time needed to execute all cases in a "test network", the required amount of budget to cover the gas requirement, and the percentage of coverage achieved for both lines and methods if we execute all available test cases. We used the test suite created and used by the developers of each project. To extract the required coverage details, we used both Hardhat (https://hardhat.org), and Solidity-coverage (https://github.com/sc-forks/solidity-coverage). Moreover, to get an estimate of the required execution time and gas for each method, we used Gas-reporter (https://github.com/cgewecke/hardhat-gas-reporter) tool.

## Experimental settings

The effectiveness of search algorithms can be greatly impacted by the settings of parameters, as noted by *Arcuri & Fraser (2013)*. It is crucial to choose the appropriate population size, stopping criterion, crossover rate, and mutation rate to prevent early convergence. In our experiments, we utilized MOEA Framework v3.2 (*Hadka, 2012*), and conducted several trials with varying population sizes of 50, 100, 250, and 500. The stopping criterion was

**Table 2** General information about the case studies.

| ID | Name | #Smart contracts | #Functions | #Lines | #Branches |
|----|------|------------------|------------|--------|-----------|
| CS1 | Chainlink | 78 | 228 | 993 | 404 |
| CS2 | PancakeSwap | 14 | 62 | 329 | 100 |
| CS3 | Venus | 146 | 1751 | 8439 | 2966 |
| CS4 | Axie | 45 | 145 | 539 | 170 |
| CS5 | Sushi | 62 | 245 | 1394 | 556 |

**Table 3** Test cases data for our case studies.

| ID | #Test cases | Execution time (ms) | Execution cost ($) | Max line coverage (%) | Max function coverage (%) |
|----|-------------|---------------------|--------------------|-----------------------|---------------------------|
| CS1 | 26 | 11,354 | 13,223.48 | 64.95 | 63.16 |
| CS2 | 17 | 3,471 | 2,745.1 | 90.88 | 90 |
| CS3 | 109 | 41,183 | 7,592.53 | 19.62 | 17.99 |
| CS4 | 10 | 34,380 | 1995.7 | 74.21 | 73.1 |
| CS5 | 35 | 8,485 | 10,342.92 | 36.66 | 37.96 |

fixed at 100k evaluations for all algorithms. For crossover and mutation, probabilities of 0.5 were used for both, per generation. We employed the Design of Experiments (DoE) approach, which is one of the most efficient and widely used methods for parameter setting in evolutionary algorithms, as outlined by *Talbi (2009)*. Each parameter was uniformly discretized into certain intervals, and values from each interval were tested for our application. Following multiple trial runs, we established the parameter values for the algorithm as 100 solutions per population and a maximum of 300 generations.

We used the MOEA Framework's standard parameter values for all other parameters. Since metaheuristic algorithms are probabilistic optimizers, they may produce different outcomes for the same problem. Therefore, we conducted 30 independent runs for each configuration and problem instance. The results were then analyzed statistically using the Wilcoxon test, as suggested by Arcuri and Fraser, with a confidence level of 95% ($\alpha = 5\%$). All experiments were carried out on a Macbook Pro machine equipped with a 2.3 GHz Intel 8-Core i9 processor, and 16 GB 2400 MHz DDR4 RAM. The Hardhat v2.6.8, Solc version: 0.8.16, solidity-coverage v0.8.2, and hardhat-gas-reporter v1.0.9 were utilized during the experiments. The Ethereum price at the time of writing the paper was $1,588, and the gas price was set at 85 GWEI.

The primary goal of comparing the mono-objective search is to determine if the three objectives are in conflict. Therefore, we assigned equal weight to all objectives after normalization between 0 and 1, for the single-objective formulation. The mono-objective approach only generates one solution, while the multi-objective algorithm generates multiple non-dominated solutions that spread across the Pareto front of the objectives. To ensure fair and meaningful comparisons, we employed the knee-point strategy to select the NSGA-II solution for the multi-objective algorithm (*Branke et al., 2004*). The knee point represents the solution with the highest trade-off between the different objectives, and it

can be considered equivalent to the mono-objective solution with equal weights for the objectives if there is no conflict among them. The knee point was chosen from the Pareto approximation by identifying the median hyper-volume IHV value. Assigning weights to objectives is often difficult as it depends on the developers' priorities and preferences. A multi-objective approach eliminates the need to assign weights to objectives, as developers can select a solution by visualizing the Pareto front based on the objectives. The knee-point method was adopted to guarantee an unbiased comparison between the two algorithms, as it is considered best practice according to the present state of computational intelligence.

## Evaluation metrics

To evaluate our approach and to answer **RQ1**, we evaluate our NSGA-II formulation against random search (RS) in terms of (1) search algorithm efficiency and (2) test optimization system performance. The aim is to confirm the necessity of an intelligent search method. This RQ serves as a basic verification and typical benchmark inquiry in any SBSE formulation effort (*Harman & Jones, 2001*). If the intelligent search method doesn't surpass random search, it indicates that the proposed formulation is inadequate.

To assess the search algorithm performance, we quantify the efficiency of each algorithm in exploring the search space. Multi-objective evolutionary algorithms, unlike mono-objective ones, produce a set of non-dominated or Pareto optimal solutions accumulated during the search process. Thus, we use three standard metrics for evaluating multi-objective optimization algorithms: Hypervolume, Spread and Generational Distance (*Zitzler et al., 2003*).

- **Hypervolume (HV):** is a performance indicator used to measure the quality of a set of solutions obtained by an algorithm. It quantifies the volume of the dominated space in the objective space, where the solution set dominates the area. A larger hypervolume value indicates that the algorithm has found a set of solutions that cover a greater area in the objective space, hence, a higher quality set of solutions. HV is used to compare the performance of different algorithms and to determine the trade-off between conflicting objectives.
- **Generational distance (GD):** is another performance indicator used in multi-objective evolutionary algorithms. It measures the average Euclidean distance between a set of solutions generated by an algorithm and a set of reference solutions, known as the Pareto front. The reference solutions are considered to be the optimal solutions, and GD provides a way to measure how close the algorithm's solutions are to the optimal solutions. A smaller GD value indicates that the algorithm's solutions are closer to the optimal solutions and hence, of higher quality. GD is used to assess the convergence of the algorithm towards the Pareto front.

In regard to **RQ2**, we defined a mono-objective algorithm that generates only one solution as output, formed by aggregating the three normalized objectives into a single fitness function. In order to measure the performance of both algorithms in solving our problem, *i.e.,* test case selection, we compare the results in terms of cost, time, and coverage

in addition to defining two more metrics:

$$Cost\_per\_FC\_point = [\frac{New\_Cost(\%)}{FC(\%)}] \tag{4}$$

$$Cost\_per\_time\_point = [\frac{New\_Cost(\%)}{Saved\_Execution\_time(\%)}] \tag{5}$$

These metrics will better illustrate the efficiency of each algorithm in maximizing the coverage while minimizing the gas cost and computational time.

**RQ3**: Our approach focuses on streamlining the test suite for the post-deployment phase, supplementing the extensive initial testing done before production. Thus, it is not anticipated for the reduced test suite to exhibit the same level of performance as the original suite, as the emphasis will be on examining the portions of the code that are more susceptible to failures due to the dynamic nature of the Ethereum network.

This RS investigates the effectiveness of the new test suite in detecting most of the critical faults? To do so, we will create various mutations to induce bugs at multiple locations. We will use the approach and mutation operators proposed by *Wu et al. (2019a)*. In summary, there are traditional mutation operators (Table 4) and Solidity specific ones (Table 5). The specific mutation operators are grouped into four groups: (1) keyword operators, (2) global variables and functions operators, (3) variable unit operators, and (4) error handling operators. For instance, functions in a contract can be declared as *view* or *pure*. A "view" function only reads data and does not modify the contract's state or emit events. "Pure" functions, on the other hand, are restricted from both reading from and modifying the state, and can only call other "pure" functions. The function state keyword change (FSC) operator can alter the behavior of a function by switching the "view" keyword to "pure," thus changing its state. A sample function state keyword change (FSC) is shown in Table 6. Another example of a mutation operator is variable type keyword replacement (VTR). Solidity is a statically and strongly typed language, meaning that the data type of each variable must be specified. An integer overflow can result in the loss of the most significant bits of the result, creating real-world security vulnerabilities. Table 7 provides an illustration of a VTR mutant, which requires testers to take negative numbers and truncation into account during testing. Further discussion about the approach, operators, and their impact on the SC are presented in *Wu et al. (2019a)*. After creating various mutations using both general and Solidity specific operators for each test case, we compare the performance of each algorithm in terms of its effectiveness in revealing faults using the following formula:

$$Effectiveness = [\frac{DetectedFaults}{TotalFaults}] * 100 \tag{6}$$

Moreover, we calculate test case fault detection rate (DR) per unit of time as follows:

$$DR = [\frac{DetectedFaults}{Time(s)}] \tag{7}$$

These metrics will demonstrate the effectiveness and efficiency of each algorithm in choosing test cases that uncover a higher percentage of severe faults.

**Table 4  General mutation operators.**

| Operator | Description |
|---|---|
| AOR | Arithmetic Operator Replacement |
| AOI | Arithmetic Operator Insertion |
| ROR | Relational Operator Replacement |
| COR | Conditional Operator Replacement |
| LOR | Logical Operator Replacement |
| ASR | Assignment Operator Replacement |
| SDL | Statement Deletion |
| RVR | Return Value Replacement |
| CSC | Condition Statement Change |

**Table 5  Specific mutation operators.**

| Type | Operator | Description |
|---|---|---|
| Keyword | FSC | Function State Keyword Change |
| | FVC | Function Visibility Keyword Change |
| | DLR | Data Location Keyword Replacement |
| | VTR | Variable Type Keyword Replacement |
| | PKD | Payable Keyword Deletion |
| | DKD | Delete Keyword Deletion |
| Global Variable and Function | GVC | Global Variable Change |
| | MFR | Mathematical Functions Replacement |
| | AVR | Address Variable Replacement |
| Variable Unit | EUR | Ether Unit Replacement |
| | TUE | Time Unit Replacement |
| Error Handling | RSD | Require Statement Deletion |
| | RSC | Require Statement Change |
| | ASD | Assert Statement Deletion |
| | ASC | Assert Statement Change |

**Table 6  Sample FSC mutant.**

| | |
|---|---|
| L1 | function allowance(address account, address spender) external pure returns (uint) { |
| L2 | return allowances[account][spender] |
| L3 | } |
| L1 | function allowance(address account, address spender) external view returns (uint) { |
| L2 | return allowances[account][spender] |
| L3 | { |

# RESULTS AND DISCUSSION

**Results for RQ1:** We evaluate the search performance of our NSGA-II based approach by comparing it to Random Search (RS). This comparison with RS is standard in introducing new search-based problem formulations to validate its efficacy. Our evaluation uses the Hypervolume (HV) and Generational Distance (GD) indicators, as described in 'Evaluation

| Table 7 | Sample VTR mutant. |
|---------|---------------------|
| L1 | function transfer(address dst, uint256 amount) external nonReentrant returns (bool) { |
| L2 | return transferTokens(msg.sender, msg.sender, dst, amount) == uint(Error.NO_ERROR) |
| L3 | } |
| L1 | function transfer(address dst, uint8 amount) external nonReentrant returns (bool) { |
| L2 | return transferTokens(msg.sender, msg.sender, dst, amount) == uint(Error.NO_ERROR) |
| L3 | } |

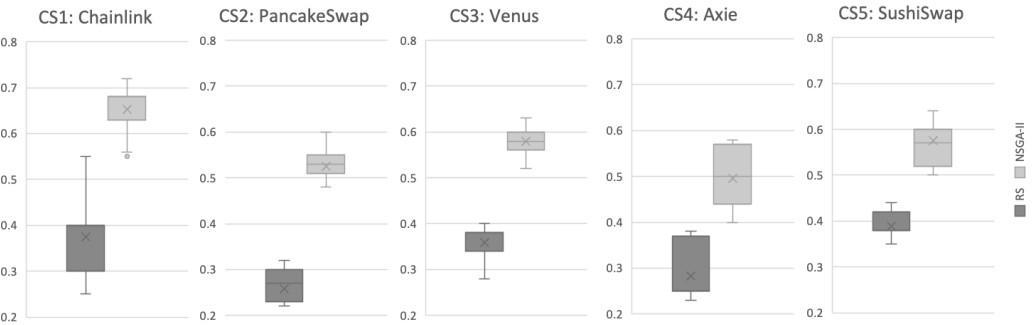

**Figure 5** Hypervolume (HV) indicator for all case studies, based on results from 31 independent algorithm runs.

metrics', over 31 independent runs for all case studies. The evaluation results are illustrated in Figs. 5 and 6. Each boxplot displays the minimum and maximum values of the indicator as the lower and upper whiskers respectively. The second and third quantiles are represented by the lower and upper boxes, the median is represented by a horizontal line dividing the boxes, and the mean value is marked by an *x*. It is evident that for the HV indicator, RS has worse values compared to NSGA-II in all case studies. To validate these results, we applied the Mann–Whitney *U* test with a 95% significance level and found a statistically significant difference between NSGA-II and RS for all case studies. For GD, the lower values, the more likely the recommendation results are better. The results indicate that NSGA-II performs well for all five case studies, while Random Search generally performs poorly (Fig. 6). This supports the conclusion that an intelligent search method is necessary for obtaining improved results in test case selection.

While the HV and GD metrics measure the efficiency of the search, we also assess the solutions found by each algorithm in terms of coverage, execution time, and cost. Table 8 summarizes the results. As discussed in Section 'Pareto-optimal Solutions', our focus is on identifying non-dominated solutions. NSGA-II is designed to optimize all objective values simultaneously, aiming for balanced results across objectives. Therefore, it's possible for RS to outperform in isolated single objectives. However, upon reviewing the results, we observe that the results of RS for all three

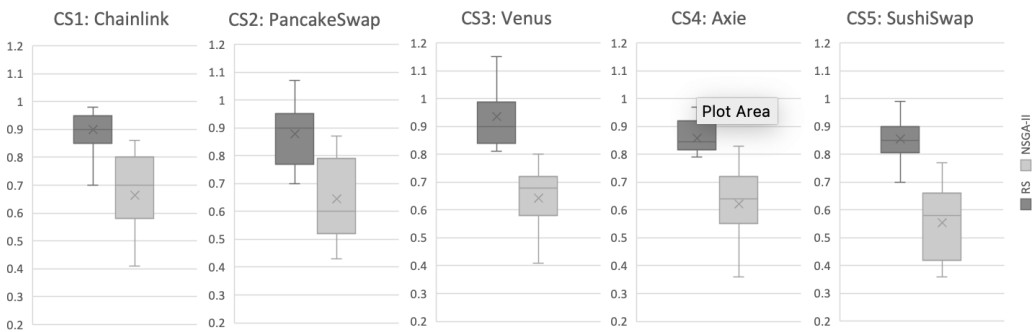

**Figure 6** Generational distance (GD) indicator for all case studies, based on results from 31 independent algorithm Runs.

objectives largely underperformed when compared to solutions from other algorithms. The only exception is CS4, where RS managed to reduce the cost and time more than the other two other algorithms. Worth mentioning that CS4 has the lowest number of test cases to select from, in addition to having the lowest total execution cost. Therefore, RS might have saved more time due to the fact that it does not require a great deal of overhead time compared to the other two, and since the selection space is relatively small, the intelligent algorithms did not have the chance to make up the difference.

However, to put that gain in perspective, we calculated two metrics as described in 'Evaluation metrics': (1) cost per coverage, and (2) cost per time saved. Tables 9 and 10 summarizes the results. From these tables, we see that RS under-performed in both metrics. In other words, the RS algorithm failed to balance between cost saving and other important metrics such as time and gas cost.

In conclusion, our use of NSGA-II in a multi-objective formulation has been proven to be effective based on a compelling evidence (addresses RQ1).

**Results for RQ2:** A mono-objective approach involves using a single fitness function composed of the three normalized objectives, producing a single refactoring solution as its output. We compare mono-objective and multi-objective algorithms. We see in Table 8 that former have better coverage in CS1, CS2, and CS4. Moreover, in CS5, mono-objective algorithm reduced the cost more than its counterpart. Again, since we are interested in all three objectives, we see in Table 9 that the multi-objective algorithm performed better in all test cases in terms of the value for money. This results confirm the fact that the three objectives we defined are actually conflicting and a multi-objective algorithms is useful to find the best trade-off between them. In Table 10, however, the mono-objective managed to outperform NSGA-II in one case study (CS5). Here, because the mono-objective covered only 7.76% of the functions, the time and cost were lower than other two algorithms. While this might be useful in some situations when the budget or time is limited, but our formulation give no preference to any particular objective and tries to maximize all in the same time. In fact, after close investigation of the code fragments covered by the mono-objective algorithm, we realized that most of the functions it covered were constructors or *pure* functions. The *pure* functions

**Table 8  Percentage of coverage, cost saving, and time saving for RS, mono-objective and multi-objective approaches.** The bold values indicate the best results.

| ID | Metric | Random search | Mono-objective | Multi-objectives |
|---|---|---|---|---|
| CS1 | LC | 44.1% | **55.09%** | 54.6% |
|  | BC | 50.24% | **56.93%** | 49.5% |
|  | FC | 51.16% | 50.44% | **54.39%** |
|  | Cost | −8.41% | −6.11% | **−11.10%** |
|  | Time | −21.03% | −13.29% | **−25.32%** |
| CS2 | LC | 35.26% | **90.88%** | 78.12% |
|  | BC | 30% | **69%** | 55% |
|  | FC | 30% | **90%** | 78% |
|  | Cost | −37.58% | −2.01% | **−60.39%** |
|  | Time | −51.05% | −2.59% | **−46.35%** |
| CS3 | LC | 12.85% | 6.23% | **12.92%** |
|  | BC | 8.87% | 5.06% | **8.29%** |
|  | FC | 12.51% | 5.71% | **13.08%** |
|  | Cost | −62.66% | −44.89% | **−72.75%** |
|  | Time | −59.08% | −51.80% | **−74.8%** |
| CS4 | LC | 41.56% | **74.21%** | 51.02% |
|  | BC | 30% | **56.47%** | 35.29% |
|  | FC | 42.06% | **73.1%** | 50.34% |
|  | Cost | **−40.96%** | 0.00% | −30.09% |
|  | Time | **−63.70%** | 0.00% | −36.29% |
| CS5 | LC | 15.85% | 8.61% | **17.43%** |
|  | BC | 9.71% | 5.58% | **11.87%** |
|  | FC | 9.38% | 7.76% | **20%** |
|  | Cost | −66.49% | **−79.62%** | −55.42% |
|  | Time | −62.02% | −76.68% | **−79.13%** |

**Table 9  Cost per function coverage point.** The bold values indicate the best results.

| ID | Random search | Mono-objective | Multi-objectives |
|---|---|---|---|
| CS1 | 2.22 | 1.86 | **1.63** |
| CS2 | 2.08 | 1.08 | **0.68** |
| CS3 | 3.27 | 9.65 | **0.64** |
| CS4 | 1.08 | 1.36 | **1.07** |
| CS5 | 3.57 | 2.62 | **2.22** |

are unable to read or change the storage used by the contract. They are employed for computing, such as in mathematical or cryptographic functions. As a result, these functions are less resource-intensive, requiring low gas and taking less time to execute. By utilizing functions that do not consume substantial computational power and gas, the mono-objective approach was able to achieve a better cost-to-time balance in the

**Table 10  Cost per time reduction point.** The bold values indicate the best results.

| ID | Random search | Mono-objective | Multi-objectives |
|---|---|---|---|
| CS1 | 4.35 | 7.06 | **3.51** |
| CS2 | 1.22 | 37.83 | **0.88** |
| CS3 | 0.67 | 1.06 | **0.09** |
| CS4 | 0.71 | 65.46 | **1.49** |
| CS5 | 0.56 | **0.26** | 0.56 |

**Table 11  Average fault revealing rate for various approaches in %.**

| ID | # of Mutations | All | Mono-objective | Multi-objectives |
|---|---|---|---|---|
| CS1 | 30 | 43.3 | 40.0 | 40.0 |
| CS2 | 30 | 56.6 | 50.0 | 46.6 |
| CS3 | 30 | 50.0 | 30.0 | 50.0 |
| CS4 | 30 | 63.3 | 53.3 | 50.0 |
| CS5 | 30 | 70.0 | 63.3 | 66.6 |

case study being considered (CS5). There is a notable high variance in the mono-objective results. This discrepancy arises from the inherent nature of mono-objective algorithms, which prioritize the optimization of a single objective function. This often leads to a limited exploration of the solution space, potentially generating solutions that are locally optimal for that particular objective. However, these solutions may lack a comprehensive or optimal balance when multiple objectives come into play. Consequently, while the outcomes may exhibit strong performance individually, such as enhanced coverage, reduced costs, or time savings, accomplishing the simultaneous fulfillment of two or more objectives becomes more challenging within the context of mono-objective algorithms.

These results show that our multi-objective approach outperforms mono-objective formulation, providing empirical proof. The conflicting nature of the three objectives supports the need for a multi-objective formulation to balance them, thereby answering RQ2.

**Results for RQ3:** An outdated test suite can lead to critical bugs and lower software quality. It's crucial to keep the test suite up-to-date with changing technology and systems. This research question investigates the impact of reducing the test suite size on bug detection. Thirty bugs were manually introduced in each case study and only code fragments covered by the initial test suite (Table 3) were mutated using 15 general and 15 Solidity-specific mutation operators (cf. 'Validation'). The findings will help determine the effectiveness and efficiency of the optimized test suite. Table 11 summarizes the results.

As expected, running all test cases gave the best detection score since it achieves the highest code coverage. We also found a clear correlation between higher coverage and higher fault revealing capabilities. What came as a surprise, however, was the bug detection score even when all test cases are executed. While some of these bugs has no effect on the execution result, many of them affected the required gas and the execution time. This

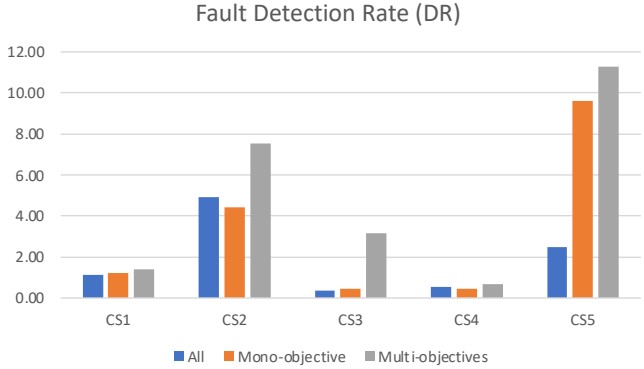

**Figure 7** **The number of faults detected per second for the algorithms.**

shows that testers pay less attention to the problems caused by solidity characteristics when writing test cases. As emphasized by *Wu et al. (2019a)*, there should be more focus on employing solidity specific mutation operators when designing test cases. An other observation was the similar score for the traditional (re-test all) and our approach in CS3, and how close they were in CS1 and CS5. The results demonstrate robust evidence of our ability to achieve reasonable fault detection rates despite significant reductions in time and cost. Note that in our formulation, all metrics were given equal importance. However, in some practical applications, the coverage may be given more weight to reveal more bugs.

When analyzing the fault detection rate in Fig. 7, it becomes clear that the traditional approach of running all test cases results in the lowest efficiency in terms of bugs detected per second. This can be attributed to the large number of test cases, some of which may be overlapping, redundant, or obsolete, which increases maintenance cost without actual value. In contrast, our proposed multi-objective formulation excels in terms of fault detection rate, as it focuses on the most effective test cases while still maintaining an acceptable level of coverage. This approach highlights the advantages of taking a multi-objective approach to testing, as it leads to improved efficiency and fault detection capabilities while avoiding the cost and limitations of running redundant or obsolete tests.

## THREATS TO VALIDITY

There are few limitations and threats to validity concerning this study. The external validity is concerned with the capacity to generalize the results. To minimize this risk, we used multiple case studies, publicly available on State of the DApps and GitHub, of different sizes, application domain, and design structure. To further validate our findings, it would be useful to replicate the study using a broader range of programs and optimization techniques. Replicating the study we conducted in this paper is part of our plan for future work. Additionally, there may be optimization algorithms or approaches to the test case selection problem that did not include in this study that could lead to improved results. Currently, there is no known algorithm that is particularly effective for solving the multi-objective test case selection problem (*Yoo & Harman, 2010*).

Another concern is related to the quality of the test cases and mutations used. The test code for all SC was obtained from the original project, and we understood that the quality of the test cases could impact our experiment. For this reason, we opted to use the test cases already established by the project developers to ensure the validity of our results. Although we used existing tools that have been used and validated in wide number of studies, there is no guarantee that the mutations realistically mimics real-world scenarios. We used, however, our experience to manually validate the test cases and mutations to minimize this risk.

The connection between what we observe and the theory lies within the scope of construct validity. To evaluate the different methods, we utilized well-known metrics like computational cost and code coverage. Additionally, we employed gas cost and fault coverage as additional metrics to compare their performance. In the future, we aim to explore different metrics and performance measures for a more comprehensive evaluation. As there is limited research in the area of SC test case selection, we compared our work to both a single-objective algorithm and the conventional retest-all approach.

There may be a threat to the internal validity of our method due to the stochastic nature of the approach and parameter tuning. We conducted various independent simulations for each problem instance to address this issue, to make sure that the multi-objective formulation is unlikely to be the only factor that influences the data. To minimize the conclusion validity threat, we used Mann–Whitney $U$-Test with 95% confidence level ($\alpha = 5\%$) and employed a popular trial-and-error approach for evolutionary algorithms (*Eiben & Smit, 2011*). We acknowledge the fact that selecting different parameters could impact the outcome. To address this, we may consider implementing an adaptive parameter tuning strategy in the future. This would involve adjusting the values during the execution to find the optimal combination for the best possible performance.

## CONCLUSION

In this study, we presented a test case selection approach for Solidity SC, which takes into account function coverage, execution time, and the gas cost required for executing the test cases. Our evaluation, conducted on various case studies, demonstrated a significant acceleration of the testing process, reduced monetary budget while still maintaining a satisfactory level of testing performance. This study aims to assist practitioners who wish to test their contracts in both test and live networks to ensure their BC applications perform as expected in real-world scenarios.

For future work, we have identified several avenues to explore. First, we aim to consider additional objectives, such as coverage of specific error-prone code fragments, security vulnerabilities, or BC-specific faults, in the search process. This will aid the testing team in managing their budget while still detecting the most severe faults that may arise due to the nature of BC networks. Secondly, we plan to investigate the integration of test generation and test selection techniques. This would enable us to automatically reduce the size of test suites when they are generated, focusing the generation effort on cases that are not already covered. Moreover, we plan to Investigating the use of machine learning techniques to

prioritize test cases, after selecting them, in a more automated and efficient manner. Finally, we plan to study the applicability of our approach to other smart contracts' programming languages, to determine the portability and generalizability of our approach.

### Funding

The authors received no funding for this work.

### Competing Interests

The authors declare there are no competing interests.

### Author Contributions

- Bader Alkhazi conceived and designed the experiments, performed the experiments, analyzed the data, performed the computation work, prepared figures and/or tables, authored or reviewed drafts of the article, and approved the final draft.
- Amin Alipour conceived and designed the experiments, authored or reviewed drafts of the article, and approved the final draft.

### Data Availability

Bader Alkhazi, & Amin Alipour. (2023). Multi-objective test selection of smart contract and blockchain applications. In PeerJ Computer Science. Zenodo. Available at https://doi.org/10.5281/zenodo.8252898

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
