# Peer review of "Multi-objective test selection of smart contract and blockchain applications"

_PeerJ Computer Science, doi:10.7717/peerj-cs.1587_

## Round 0.1 · original submission · Minor Revisions

I concur with the reviewers' recommendation. Please address the list of minor issues and resubmit.

Reviewer 1 ·

Basic reporting

In the beginning, the paper shows that several features can be offered by blockchain technology, such as immutability, decentralization, transparency, and security (offering the existing solutions/references that blockchain technology has been successfully applied in healthcare, manufacturing, governance, and the Maesa and Mori domains, according to recent studies). The "Background and Related Work" section provides the contexts of the current blockchain technology, the precise definition of smart contracts and its working mechanism. In addition, the manuscript is clearly written in professional, unambiguous language.

Experimental design

The paper's overall design considers one important phenomenon: the performance testing of blockchain applications can sometimes involve both controlled and real-world environments. As different blockchains use different programming languages and are deployed in various virtual environments, determining how they should interact can be extremely challenging. Figure 1 illustrates the overall idea of the proposed approach clearly. The "experimental settings" section shows that this paper defines a set of reasonable parameter values to achieve the high effectiveness of the testing. To ensure fair and meaningful comparisons, the paper employs the knee-point strategy to select the NSGA-II solution for the multi-objective algorithm.

Validity of the findings

An analysis of five real-world Solidity projects State of the DApps1 and GitHub is presented in this paper to evaluate the effectiveness of the proposed approach. The paper compares the effectiveness of the proposed approach against other common test selection techniques, including mono-objective test selection and baseline random selection. One major research finding is that this is the first research work targeting the smart contracts test case selection and uses a multi-objective algorithm to select the test cases in order to minimize gas cost and execution time while maximizing the coverage. The validity of the findings is enhanced by comparing the effectiveness of the proposed approach with the state-of-the-art approaches based on real-world Solidity projects. The Q&A style in the Validation section is impressive.

Additional comments

The introduction of "Software Testing" needs more detail. I suggest that you improve the description and add more references about white box testing and black box testing.

The paper states that you use the bit-stream mutation operator to randomly select one test case in the vector dimension in order to replace it with another entry. Besides the bit-stream mutation operator, are there any other ways can achieve this? If so, why did you pick the bit-stream mutation operator?

Please enlarge your case studies scope to show that the proposed approach has real contributions to the community. Like, considering the top 10 blockchain platforms, written in Solidity, and has its open-source code for the smart contract published.

minor: please only define the corresponding short name when the term first appears, and use this defined short name consistently throughout the paper (for example, "blockchain (BC) applications" is defined multiple times, and using BC technology consistently once you defined it). Minor typos as well, such as "effectivenss".

Cite this review as

Reviewer 2 ·

Basic reporting

Overall the paper is well written, with sufficient references to related researches and detailed explanation of the method and experimentation.

But there's one important detail that I failed to understand. The paper proposes a method to select test cases for smart contract and blockchain applications, using multiple objectives and fitness functions. The paper says that a multi-objective algorithm is useful to find the best trade-off between them. But I failed to understand from the paper what's the definition of the "best" trade-off and how to find it? The objectives can be conflicting with each other, so how to make the trade-off between them? For example when two objectives are conflicting with each other, is it possible that in some cases we want to favor one of the objectives when making the trade-off while in other cases we want to favor the other objective? How it is done with 3 conflicting objectives? The trade-offs between 3 conflicting objectives seem to be more complicated than 2 conflicting objectives.

I recommend to revise the paper to explain this better.

Experimental design

No comment.

Validity of the findings

The main contribution/novelty of the paper is that it is the first research of test case selection for smart contract and blockchain applications. It seems to me that the main challenge of test case selection for smart contract and blockchain applications is that it needs to consider more objectives (test coverage, execution time and execution cost ) , compared to many other applications for which only one or two objectives are needed. I wonder if there is any other domain that needs to consider more objectives for test case selection? If so, the paper's proposed method can also be applied to these domains? Or if there is any existing research on test case selection with multiple objectives but for other domains/applications?

Cite this review as

Reviewer 3 ·

Basic reporting

The writing of the manuscript needs improvement, there are multiple grammar and syntax errors in it. Figures and tables could be also reallocated to further the readability of the draft. Last, some acronyms are never defined.

Experimental design

The specifics of the NSGA-II algorithm are not explained and it would be best to disclose the testing benchmarks, e.g., in a git repo. Additionally, the authors state "a set of reasonable parameter values
was experimented with" but do not identify these parameters. Last, I am confused as to why they use gas-reporter instead of running the experiments directly to measure the SC costs.

Validity of the findings

There are some inconsistencies between the data reported in the tables/figures and in-text. The authors state in line 575 that "RS under-performed in all case", which is clearly contradicted by tables 8 & 10.
I suggest you explain also why the mono-objective method shows such variance in the results in table 8.

Additional comments

Overall, I find this approach rather limiting, in the sense that you are exploring only Ethereum smart contracts. I would suggest you execute your methods for other types of smart contracts as well and also motivate further (with specific examples maybe) cases where the live testing is more meaningful and thus complimentary to the testnet one.

Cite this review as

---

## Round 0.2 · accepted · Accept

The authors have sincerely addressed all the reviewers' comments. I think it is ready to be accepted.